# Economic Sustainability of Small-Scale Hydroelectric Plants on a National Scale—The Italian Case Study



**Anita Raimondi** [1,*] , **Filippo Bettoni** [2] , **Alberto Bianchi** [1] and **Gianfranco Becciu** [1]

[1] Dipartimento di Ingegneria Civile e Ambientale, Politecnico di Milano, 20133 Milano, Italy; alberto.bianchi@polimi.it (A.B.); gianfranco.becciu@polimi.it (G.B.)
[2] Independent Researcher, 24020 Azzone, Italy; filippo.bet@icloud.com
[*] Correspondence: anita.raimondi@polimi.it

**Abstract:** The feasibility of hydroelectric plants depends on a variety of factors: water resource regime, geographical, geological and environmental context, available technology, construction cost, and economic value of the energy produced. Choices for the building or renewal of hydroelectric plants should be based on a forecast of the future trend of these factors at least during the projected lifespan of the system. In focusing on the economic value of the energy produced, this paper examines its influence on the feasibility of hydroelectric plants. This analysis, referred to as the Italian case, is based on three different phases: (i) the economic sustainability of small-scale hydroelectric plants under a minimum price guaranteed to the hydroelectric operator; (ii) an estimate of the incentives for reaching the thresholds of "acceptability" and "bankability" of the investment; (iii) an analysis of the results obtained in the previous phases using a model of the evolution of the electricity price over the 2014–2100 period. With reference to the Italian case, the analysis suggests that, to maintain the attractiveness of the sector, it is necessary to safeguard the access to a minimum guaranteed price. With the current tariff plan, complete sustainability is only achieved for plants with $p \leq 100$ kW. For the remaining sizes, investments under current conditions would not be profitable. The extension of minimum guaranteed prices could make new medium-large plants (500–1000 kW) more attractive. The current incentive policy is not effective for the development of plants larger than 250 kW, as systems with lower capital expenditures are preferred. Uncertainty about the evolution of the price of energy over time is a concern for the sector; the use of evolutionary models of technical economic analysis tried to reduce these criticalities, and it was shown that they can be transformed into opportunities. It was also found that profitability due to the growing trend expected for the price of energy cannot be highlighted by a traditional analysis.

**Keywords:** economic sustainability; mini hydroelectric plants; tariff; incentive; climate change

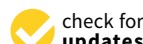



## 1. Introduction

Hydropower accounts for about 20% of worldwide electrical power production, with a higher percentage in mountain regions [1]. It is a clean source of energy as well as an economic resource for regions rich in usable water. Hydroelectric production is managed mainly according to water availability and the selling price of electricity [2]. Electricity demand and price generally depend on societal and economic development, but they are also subject to changes related to weather variability, of both a seasonal and long-term type (e.g., variation due to climate change [3,4]). Water availability depends largely on climatic and hydrological conditions and, therefore, can have significant variations in both space and time [5].

While seasonal and year-to-year variability of river runoff has always been taken into account in the past, the recent concern is related to the effect of climate change on the productivity of hydroelectric plants. This effect is related not only to precipitation volumes and time patterns, but also to evapotranspiration and consequently to average

temperatures. While the productivity of reservoir hydroelectric plants is mainly affected by variations of the available water volume (i.e., by the annual rainfall volumes), in the case of run-of-the-river hydroelectric plants it is also affected by the time distribution among seasons of precipitation, that is, mainly by the river discharge-duration curve [6].

Hydropower often comes largely from cold water, originating from ice/snow melting in the mountain areas, subject to rapid cryosphere wasting due to global warming [7]. This could be a concern, especially for the Alps, with a large share of hydropower depending on cryosphere water. In the Alpine region, the rising temperatures resulted in the loss of more than half of the volume of glaciers since 1900. With a global temperature increase of 2–4 degrees, 50–90% of the ice mass of the mountain glaciers could disappear by the end of this century [8]. With earlier snow melting and rainfall variation, inter-annual run-off is changing towards less water during summer and more during the winter season. Depending on the watershed, the quantity of water may increase initially due to the loss of ice stock [9]. Changes in temperatures and precipitation patterns can have profound effects on water systems and cause important changes on uses that are highly dependent on the hydrological regime, such as hydropower production, in turn modifying total annual inflow volumes and their seasonal distribution [10].

Several studies analyzed climate change impacts on hydropower production. Finger et al. (2012) described how these changes affected water resources and subsequently hydropower production in hydropower plants in a glacierized alpine valley (Vispa valley, Switzerland). The trends observed in all the projections indicated significant changes to the current situation: the future melt- and rainfall-runoff will increase during spring but decline during summer [1]. Stucchi et al. (2019) studied the climate change impact on the Sabbione (Hosandorn) glacier, in the Piedmont region of Italy, and the homonymous reservoir, which collects water from ice melt; they projected the hydrological cycle under properly downscaled climate change scenarios until 2100. They concluded that the decrease of cold water in this area, which is paradigmatic of the present state of hydropower in the Alps, and the subsequent considerable hydropower losses due to climate change call for adaptation measures [7]. Ravazzani et al. (2016) assessed the impacts of climate change on hydropower production of the Toce Alpine river basin in Italy; they showed an increase in production in autumn, winter, and spring, and a reduction in June and July [11]. Gaudard et al. (2014) provided a synthesis and a comparison of methodologies and results obtained in several studies devoted to the impact of climate change on hydropower in the Swiss and Italian Alps [12]. Duratorre et al. (2020) studied the effects of potential climate change scenarios at 2100 on hydropower production from the Chavonne plant, in the Valle d'Aosta region of Italy, using Poli-Hydro, a state-of-the-art hydrological model to mimic the hydrological budget of the area, including the ice and snow melt share [13]. Patro et al. (2018) assessed the impacts of nine climate-change scenarios on the hydrological regime and on hydropower production of forty-two glacierized basins across the Italian Alps, for the period 2016–2065. Results predicted a decline in average summer runoff across all basins compared to present levels, due to the glacier shrinkage, whereas different temperature or precipitation trends play a marginal role [14].

In areas where a large share of hydropower production depends on ice melt, the expected future lack of water due to the reduction of glaciers may affect energy production and requires adaptation strategies [15]. Sensibility of the productivity of hydropower plants to climate change, together with the fact that they are still a major source of renewable energy, makes supporting and increasing this type of energy production and strengthening the efforts to reduce the human induced climatic changes strategic [16].

The focus of this paper is the economic sustainability of small-scale hydroelectric plants on a national scale. Energy and climate policy, as well as electricity market design and dynamics, plays a pivotal role for the future of the sector [17,18]. Among the Italian renewable sources, according to the GSE's (Energy Services Operator) statistical report, hydroelectricity is the largest source, in terms of both power and annual production. It was proven to be the most efficient source because with the same installed power and incentive

paid, it produces more energy than other sources, since its useful life is much longer. Hydroelectric plants have a much higher investment cost than other energy production technologies, but operating costs are a lot lower since no type of fuel is required, which is often the most important cost component. The relationship between the energy produced during the useful life and the energy consumed to build, install, and dispose of it at the end of its life is of an order of magnitude higher than that for other types of renewable sources. Moreover, this relationship ensures the security of the energy supply since hydroelectricity is the only programmable renewable source; reservoir plants produce only when necessary, thus stabilizing the National Transport Network and allowing the Italian electricity system to adapt to our consumption hour by hour. Even run-of-the river plants have a predictable production in the short term, and therefore they have a qualitatively better role in the energy system than wind power or photovoltaic plants. In any case, there are two main critical issues that undermine the economy of an investment in hydroelectric plants: the cost of water and the lack of economies of scale since the plants are tailor-made for their respective sites. The design of small-scale hydroelectric plants is a challenge involving several factors: hydrological, technological, environmental, and social. Moreover, each plant must undergo to a strict and selective authorization process, facing the regulatory uncertainties regarding possible incentives.

The objective of the following study is to analyze the economic sustainability of small-scale river hydroelectric plants with a power concession of up to 1 MW in the current regulatory context, in order to provide planners, legislators, and stakeholders with reflections that are useful for the transition to a new configuration of incentive mechanisms. The choice of focusing on small-scale hydroelectric plants is due to the peculiarity of the territory. The Alps and the Apennines are completely saturated, and it is impossible to build large systems.

## 2. Materials and Methods

The study is divided into three phases. The first is an analysis of the economic sustainability of hydroelectric plants with power concession up to 1000 kW in the absence of incentives for the first 1,500,000 kWh produced. Particular attention is paid to the incidence of water concession fees on the economic evaluation of the investment. The second phase is an estimate of the value of the incentive needed to achieve economic sustainability for hydroelectric plants, compared to the investment "acceptability" and "bankability" thresholds typical for these types of plants. The last phase consists of an evaluation of the sustainability of the plants in the complicated context of climate change, with reference to the most influential factors that govern the phenomenon in such a way as to offer the most reliable and truthful forecast possible. The aim is to understand whether the incentives can be seen to be a shock absorber capable of effectively meeting the economic needs of the hydroelectric sector, in order to ensure that the latter remain strategic in the Italian production system.

To evaluate the economic suitability of small-scale hydroelectric plants, two levels of feasibility were adopted:

(a) Economic feasibility: This is determined by the profitability rate, that is, the internal rate of return (IRR), between 7% and 9%. This represents a profitability range typically considered acceptable by the entity that promotes the investment.

(b) Banking feasibility: Rates higher than 9% on average are considered acceptable by credit institutions to guarantee the bankability of a hydroelectric project.

An economic analysis of the suitability of small hydroelectric plants can be conducted using different methods. The simplest is to compare the relationship between the total investment and the installed power or the ratio between total investment and annual energy yield. These criteria do not identify the value for money of the systems since the revenues are not considered; they can only be used to obtain general indications on the investment. In this study, the net present value (*NPV*) methodology was used. This makes it possible to obtain a faithful estimate of the profitability of the project by estimating the

IRR. The *NPV* is simply the difference between cash inflows and outflows, throughout the duration of the investment, both discounted at a rate called the discount rate [19–21]. The formula to calculate the *NPV* [22], given the condition that the cash flows occur at regular time intervals, is:

$$NPV = \sum_{i=0}^{n} \frac{R_i - (I_i + O_i + M_i)}{(1+r)^i} + V_r \tag{1}$$

where $I_i$ is capital expenditures in period $i$, $R_i$ is cash inflow in period $i$, $O_i$ is operating cash costs in period $i$, $M_i$ is maintenance and repair cash costs in period $i$, and $V_r$ is current residual value of the investment at the end of its lifetime, $r$ is the discount rate or opportunity cost of capital, and $n$ is the number of periods considered.

Only projects with positive *NPV*s can be considered acceptable. The IRR indicates the rate of return expected from an investment: the higher the IRR is, the better value-for-money the investment represents. The limit condition is as follows:

$$NPV = f(r = IRR) = 0 \tag{2}$$

**3. Case Study**

In Italy there are 3700 hydroelectric plants, and they achieve a total production of 42.4 TWh, which is about 14% of the country's production [19]. Most such plants are small and have marginal influence. In 2015, 77% of Italian hydropower was produced by plants with power >10 MW, covering merely 17% of the total electricity production. The distribution of hydroelectric plants is quite heterogeneous, as well as the density of national installed power. Figure 1 shows a map of Italian maximum annual hydroelectric production. The largest number of plants in Italy is in the northern regions with very high percentages in Piedmont, Lombardy, and Trentino Alto-Adige. According to data provided by the GSE (year 2016), over 55% of all the plants are installed in these three regions alone, with a considerable density in Piedmont and Lombardy where there are 36% of all the plants installed, which generate a total of 42% of the national installed hydroelectric power [19]. Lombardy was chosen as a case study, given the high presence of hydroelectric plants and because it is a virtuous example in terms of the canons of concession for small plants. Consequently, if the results were not advantageous for these types of plants and in this context where the ferment and private initiative are masters, they would not be advantageous in other regions either, where the overall rents are higher. In the calculation, the following standard sizes were used: 100, 250, 500, and 1000 kW, all of which are in the category of microhydroelectric plants.

*3.1. The Italian Tariff System*

One of the main concerns regarding the economic sustainability of hydroelectric plants in Italy is that while the average energy price is constantly decreasing (i.e., about 20% less than that in 2012), the sum of the water concession fees/surcharges continues to rise; in some regions there was an increase in taxes related to the use of water resources by local authorities by almost 160% [23]. Figure 2 shows the growing trend of water concession fees in recent years for the two regions "at the extreme" as regards the values for small plants in comparison to the national single price (PUN). After the peak of 2012, the PUN went down (−17% overall from 2009 to 2018), while the water concession fees, especially in Piedmont, increased significantly (+159% from 2009 to 2018). The increase in Lombardy is less marked and more gradual (+12% in the period 2009–2019). The trend is rather disconcerting, considering that the fees are fixed costs for the hydroelectric operator and are not related to either the economic value of production or actual water availability [23].

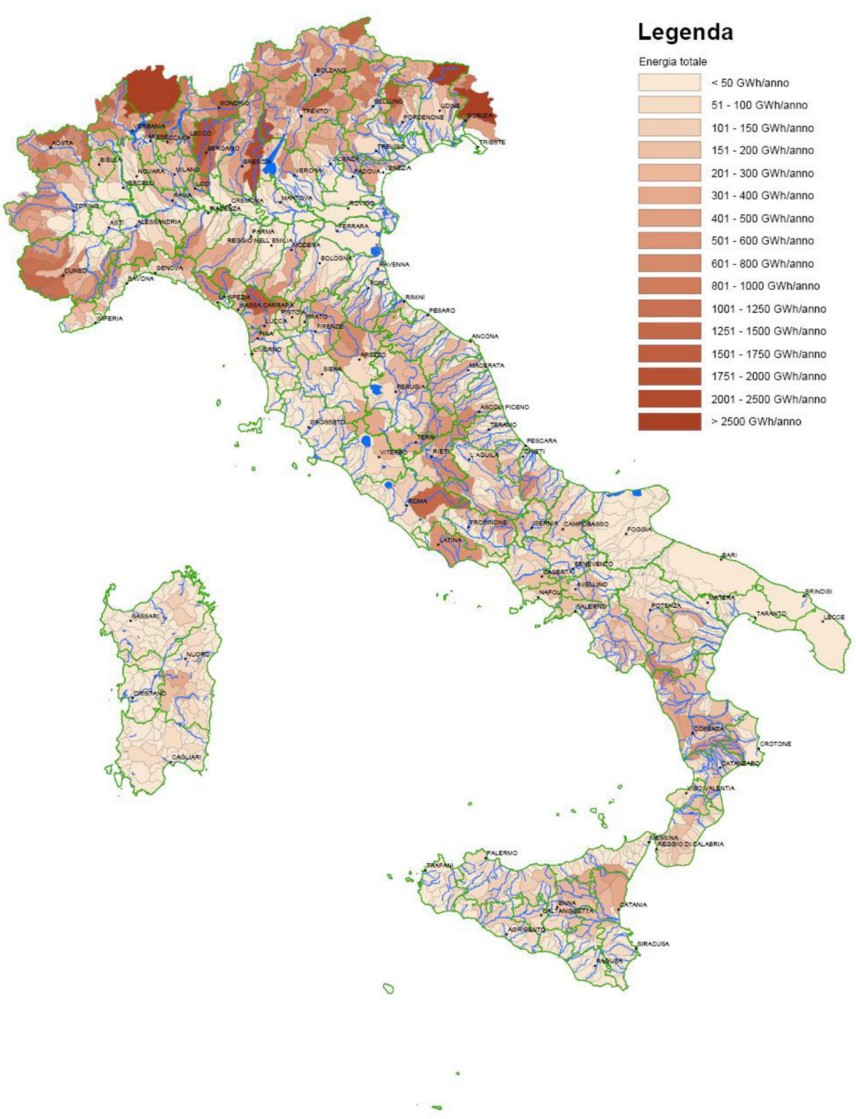

**Figure 1.** Map of maximum annual hydroelectric production in Italy (Source: ERSE SpA, https: //www.erseambiente.it/, accessed on 15 April 2021).

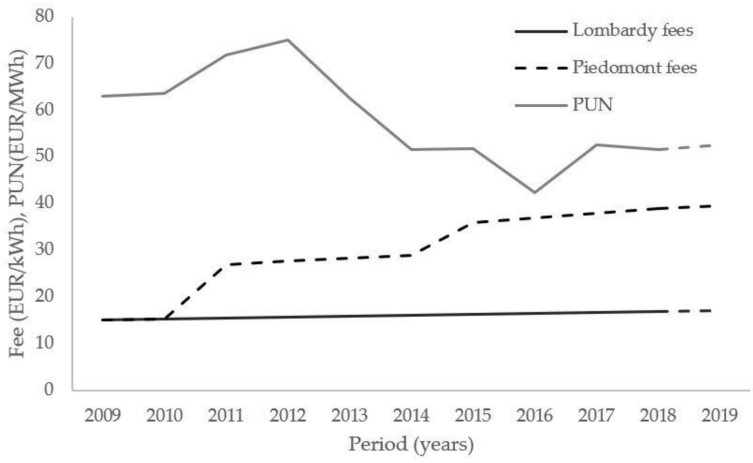

**Figure 2.** Water concession fees in the period 2009–2019 [23].

The water concession fee varies according to the region, and at above a predefined threshold (220 kW), it is necessary to pay surcharges as defined by a national standard. Table 1 shows an example of annual water fees for Lombardy.

**Table 1.** Annual water fees for Lombardy [23].

| Annual Water Fees In Lombardy | |
| --- | --- |
| Water concession fee | 16.19 EUR/kW |
| Extra charge for mountain watershed | 30.67 EUR/kW |
| Extra charge for local authorities | 5.78 EUR/kW |
| Ichthyogenic fee | 0.85 EUR/kW |
| Royalties | 3% on revenues |

The investment in a small hydroelectric plant involves several payments distributed over the life of the project and provides incomes, also distributed over time. Outputs include a fixed component such as the cost of capital, insurance, taxes other than income taxes, etc., and a variable component represented by operating expenses and maintenance, which are costs that absolutely cannot be ignored for a correct assessment of economic profitability and above all with a view to efficient operation throughout its useful life. At the end of the project, which generally coincides with the duration of the concession, the residual value should be positive. The sale price of the energy produced is defined through the so-called PMGs (minimum guaranteed prices) or according to a simplified tariff mechanism that allows producers to sell the electricity fed into the grid, transferring it directly to the GSE who remunerate them for it based on precise and variable rates every year, paying a price for each kilowatt hour drawn. In this economic model, it is assumed that the PMGs are constant for the entire duration of the plant's concession. The choice is dictated firstly by the awareness that in the last ten years these rates have remained virtually unchanged and secondly by the desire to recognize them as being of more and more strategic importance to support the sector; in this way, the study can constitute a well-founded alternative for a possible future proposal of legislation devoted to environmental protection. Tables 2 and 3 show PMGs until 2013 and current PMGs (2019) respectively.

**Table 2.** Minimum guaranteed prices in 2013 [23].

| Guaranteed Minimum Prices (EUR/mwh) | 2013 |
| --- | --- |
| 0–250,000 (kWh) | 158.7 |
| 250,000–500,000 (kWh) | 100.5 |
| 500,000–1,000,000 (kWh) | 86.7 |
| 1,000,000–1,500,000 (kWh) | 80.6 |

**Table 3.** Minimum guaranteed prices in 2019 [23].

| Guaranteed Minimum Prices (EUR/MWh) | 2019 |
| --- | --- |
| 0–250,000 (kWh) | 156.1 |
| 250,000–500,000 (kWh) | 107.2 |
| 500,000–1,000,000 (kWh) | 67.7 |
| 1,000,000–1,500,000 (kWh) | 58.5 |

Table 4 shows the incentive plans proposed in recent years for flowing water systems; in the last decade, the incentive has dropped by about 30%.

**Table 4.** Tariff associated with different power for different ministerial decrees [24–26].

| Flowing Water | Power (kW) | Rate (EUR/MWh) |
|---|---|---|
| DM 6 July 2012 | $1 < p \leq 20$ | 257 |
| | $20 < p \leq 500$ | 219 |
| | $500 < p \leq 1000$ | 155 |
| DM 23 June 2016 | $1 < p \leq 250$ | 210 |
| | $250 < p \leq 500$ | 195 |
| | $500 < p \leq 1000$ | 150 |
| DM 4 July 2019 | $1 < p \leq 400$ | 155 |
| | $400 < p < 1000$ | 110 |
| | $p \geq 1000$ | 80 |

The incentive tariffs provided for medium-large sized plants underwent a percentage decrease in the incentive tariffs higher than that for all the other sizes.

*3.2. The Economic Value of Energy Produced*

Several models aimed at estimating the electricity prices were recently introduced in Europe in the wake of the liberalization of the energy market at the end of the 1990s. Such liberalization led to the country-wise definition of the free market [27]. In Italy, Legislative Decree 79/1999 allowed for such liberalization. Energy prices set after the energy liberalization are made available continuously by the Energy Markets Operator (GME), an Italian authority with the mission of promoting the development of a national competitive electricity system, according to the criteria of neutrality, transparency, and objectivity. Competition in the electricity market is guaranteed by the Borsa Elettrica, an electricity stock market. It promotes the application of efficient equilibrium prices, allowing for the sale and purchase of electricity based on greater economic convenience. It is organized as a real physical market, with the definition of sales and purchases through hourly charts, according to the criterion of economic merit. This consists of considering the prices in increasing order for sales and the prices in decreasing order for purchases. Price definition takes place as in a physical market, by matching supply and demand. Electricity offers are accepted in order of economic merit, i.e., in order of increasing price, until their sum in terms of kWh completely meets the demand. The kWh price of the last accepted bidder, i.e., the one with the highest price, is attributed to all offers, and according to European Directive 2009/28, renewable energies, such as hydropower, have priority in terms of access to the market. In so doing, in each zone of the Italian territory with given technical constraints, the equilibrium prices are defined, i.e., those that are found at the intersection of the supply and demand curves. Subsequently, the PUN is established by GME. The economic value of electricity is difficult to express by means of a relationship between the independent variables, even only at a national level. These can be physical, economic, social, and political variables and are therefore all specific to a sociopolitical context, generally referable to a national scale. In literature, some models were proposed for the economic value of electricity linked to more general quantities that are useful if future projections are to be made [28,29]; among them there are multi-agent models [30], parametric models [31], stochastic models [32–34], and computational models [35]. In addition, hybrid, or mixed, models have also been developed [3,4,36–39]. The availability of water resources in the coming years, strongly influenced by climatic changes [28], is expected to greatly influence hydroelectric production and, consequently, the economic value of energy produced. Moreover, hydroelectric energy is greatly affected by weather conditions; its productivity can be subject to significant seasonal and annual variations. The climatic conditions affect the hydrological cycle, the energy demand, and the price of electricity. Bombelli et al. [3,4] investigated how hydroelectric production is influenced by the climate, the fluctuations in demand, and price constraints, and in doing so they extrapolated a hypothetical trend of electricity price up to the end of the 21st century, from three global climate models of the IPCC AR5, RCP2.6, RCP4.5, and RCP8.5 (Figure 3). The

method was applied to the case study of the Italian electricity market, showing acceptable capacity for modelling recently observed price fluctuations.

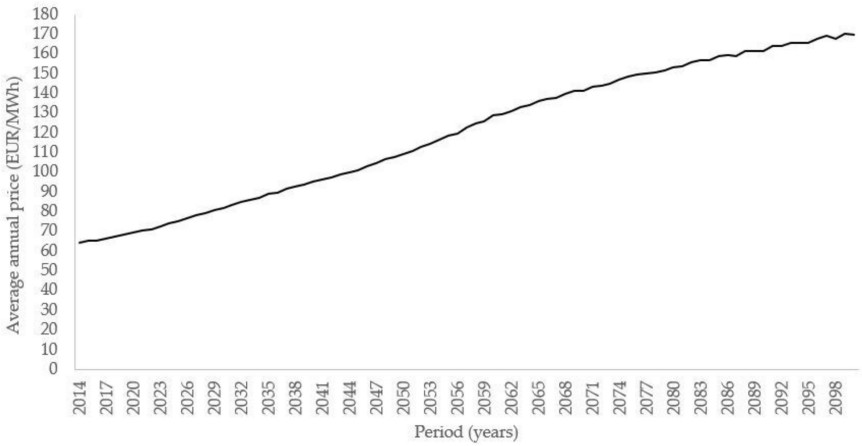

**Figure 3.** Trend of average annual prices of electricity over the period 2014–2100 [3,4].

The average annual price of energy is expected to undergo a significant increase over the 87 years analyzed, going from around 64 EUR/MWh to around 169 EUR/MWh. The immediate outcome of such a projection is the increase in expectations of the entire hydroelectric sector. If currently the economic sustainability of a hydroelectric project cannot be separated from incentive policies, such a scenario may instead reserve the possibility of investing in the sector without the need to rely on subsidized tariffs.

*3.3. Phase 1*

In phase 1 the economic sustainability of hydroelectric plants with concession power up to 1000 kW in the absence of incentives and access to the PMG for the first 1,500,000 kWh produced was analyzed. The energy exceeding the PMG threshold is sold at the market price, that is, the average value of the last 5 years of the PUN on the day before market, was assumed. Moreover, particular attention was paid to the incidence of water concession fees on the economic evaluation of the investment. In calculating, energy was weighted by a reduction coefficient equal to 0.85 to consider that a plant is not always at its maximum potential due to periods of inactivity or other external factors that the operator cannot exclude. These could be periods of drought (in which the plant runs at reduced power) or periods of full extremes (in which it runs at maximum power or is stopped). From this point of view, the resulting economic simulation of the profitability of a plant is certainly more reliable and representative of reality because it considers the unpredictability. Furthermore, according to this logic, the average hours of operation considered are effective (nonoperating hours for maintenance excluded). The calculation is usually carried out over 30 years because due to the discounting, both expenses and income weigh shortly after many years. This aspect, which could be considered a "limit" of this economic model, is not, however, influential in this analysis since it is customary to consider a duration equal to the period of concession of the plants, which in the greatest number of cases is around 20–30 years.

Figure 4 shows the trend of the IRR varying the average annual hours of operation between 3000 and 8000 h and the CAPEX (capital expenditure) between 4000 and 7000 EUR/kW for sizes of 100, 250, and 500 kW, and between 3000 and 6000 EUR/kW for size 1000 kW. Average OPEX (operating expenses) of 125 EUR/kW for the size of 100 kW and of 115 EUR/kW for the others and an annual inflation rate of 1% were assumed. In each graph, two areas were highlighted to mark the investment acceptability threshold in orange (for profitability rates between 7% and 9%) and the investment convenience threshold in green (for rates higher than 9%). Although the average operation of the entire national hydroelectric park, including the reservoir plants, is equal to 3370 h/year, the

plants analyzed are generally characterized by greater hours of operation; this is because little run-of-river systems, which guarantee a more persistent functioning throughout the year that easily reaches 6000–7000 h/year, were considered.

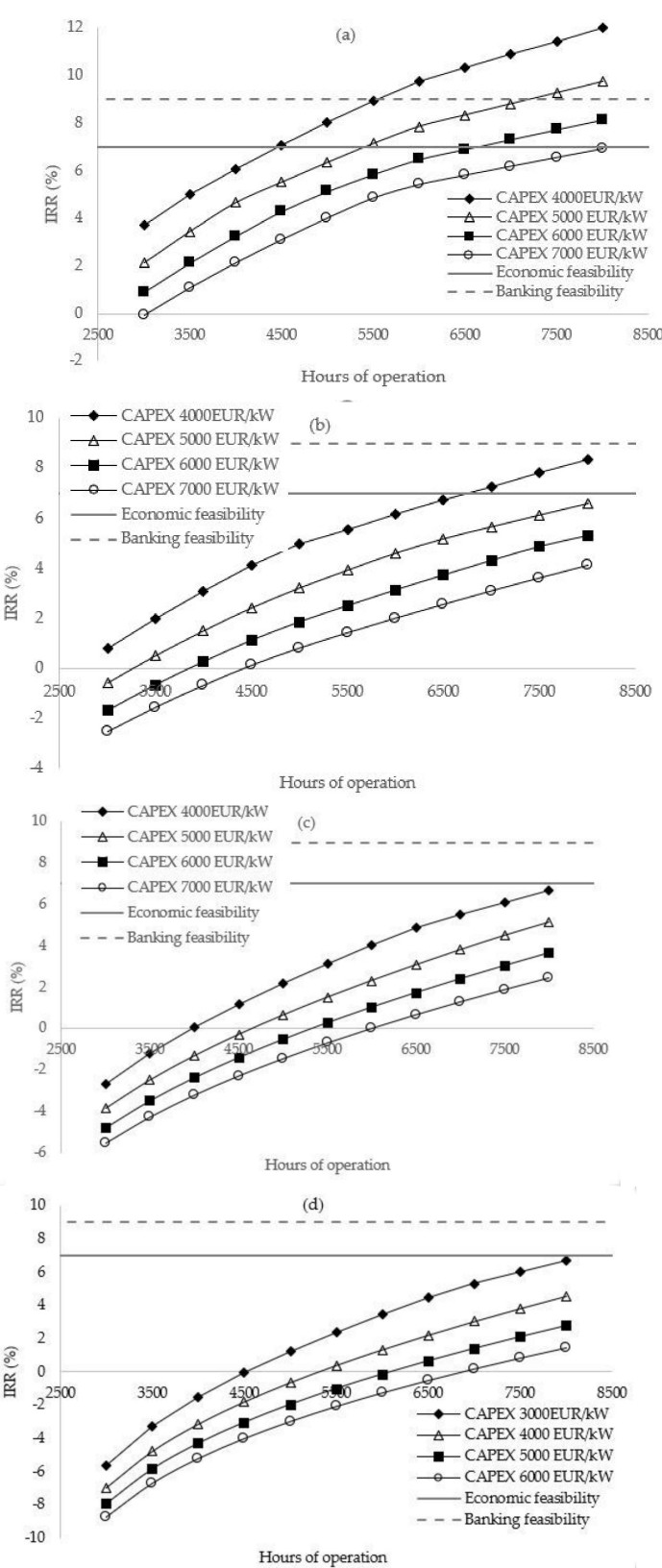

**Figure 4.** Internal rate of return vs. hours of operation for different capital expenditures for systems of 100 kW (**a**), 250 kW (**b**), 500 kW (**c**), and 1000 kW (**d**).

The IRR of the investment is less than 7% for almost all cases, except the first. The rate worsening as the size increases is substantially due to the lesser relevance of the PMG provided for plants with a concession power less than 1000 kW, within the first 1,500,000 kWh produced. Only the smaller plants can take advantage of these subsidized prices for all the produced energy, while the larger ones must operate mainly on the market price. As shown in Figure 4, a 100-kW system is the only interesting one: it achieves a threshold of acceptability for a CAPEX of 4000 EUR/kW, 5000 EUR/kW and 6000 EUR/kW respectively after 4500, 5500 and 6500 h. For plants with an installed power of 250 kW, the area of acceptability is only slightly crossed with a CAPEX of 4000 EUR/kW. Systems with the most expensive installation costs (CAPEX equal to 6000 and 7000 EUR/kW) achieve a very low IRR. When a plant exceeds 5,000,000 kWh of energy production, according to PMG 2019, the price of energy undergoes a sharp decrease of approximately 37%, from 107.2 to 67.6 EUR/MWh. The cash flow is therefore gradually reduced as the production and consequently also the IRR increase. As shown in Figure 4c,d, the situation worsens for plants of 500 and 1000 kW, as these can reach very high productions, of about 3,000,000 and 7,000,000 kWh, respectively. In such cases, the effort to use substantial resources for development is not rewarded at all by the high productions because the subsidized tariff plan does not reward them. The simulation concerning the 1000 kW system has the lowest CAPEX, i.e., between 3000 and 6000 EUR/kW, in order to consider the scale factor; unit installation costs that are too high would lead to a hugely unreliable investment. The rents (in which royalties are also considered) have a strong impact on the profitability of the plant: these represent an average annual cost equal to approximately 6% of the revenues for a 100 kW plant, also exempt from the payment of mountain watershed and local authorities, between 9% and 12% for a 250 kW plant, between 15% and 17% for the 500 kW case, and between 19% and 22% for a 1000 kW system. If incentive tariffs on the same cases were applied, any plant size would become economically sustainable, even hitting profitability peaks of over 25%. Figure 5 shows results for the following incentives: 0.219 EUR/kWh for 250 kW plants (Figure 5a), 0.179 EUR/kWh for 500 kW plants (Figure 5b), and 0.1561 EUR/kWh for 1000 kW plants (Figure 5c). To consider the scale factor, the rate was discounted as the size increases. Despite this, all lines intersect the areas of acceptability and convenience.

In Figure 5, only the minimum and maximum CAPEX were represented. The IRR achieves the profitability threshold for all combinations even for the highest CAPEXs. This proves that, in a regulatory context that supports the sector with advantageous tariffs, there could be significant sustainable development rates for all plant sizes without further concerns about investment attractiveness. Newly built plants to which the PMGs are applied do not appear to be economically sustainable, according to the current thresholds. They were extended to up to 2,000,000 kWh to highlight the positive influence produced by GMPs, especially for high hours of operation. Given the low price guaranteed for the last bracket, i.e., between 1,000,000 kWh and 1,500,000 kWh, instead of increasing the latter, it was decided to add the 500,000 additional kWh that is missing to reach 2,000,000 kWh in equal measure (i.e., 250,000 kWh each) in the two central brackets. For all plant sizes, the curves at CAPEX 4000 EUR/kW were compared with two alternative tariff plans: the tariff plans of PMGs 2013, which show more favorable prices, and PMGs 2019, with an extended PMG to 2,000,000 kWh (Figure 6).

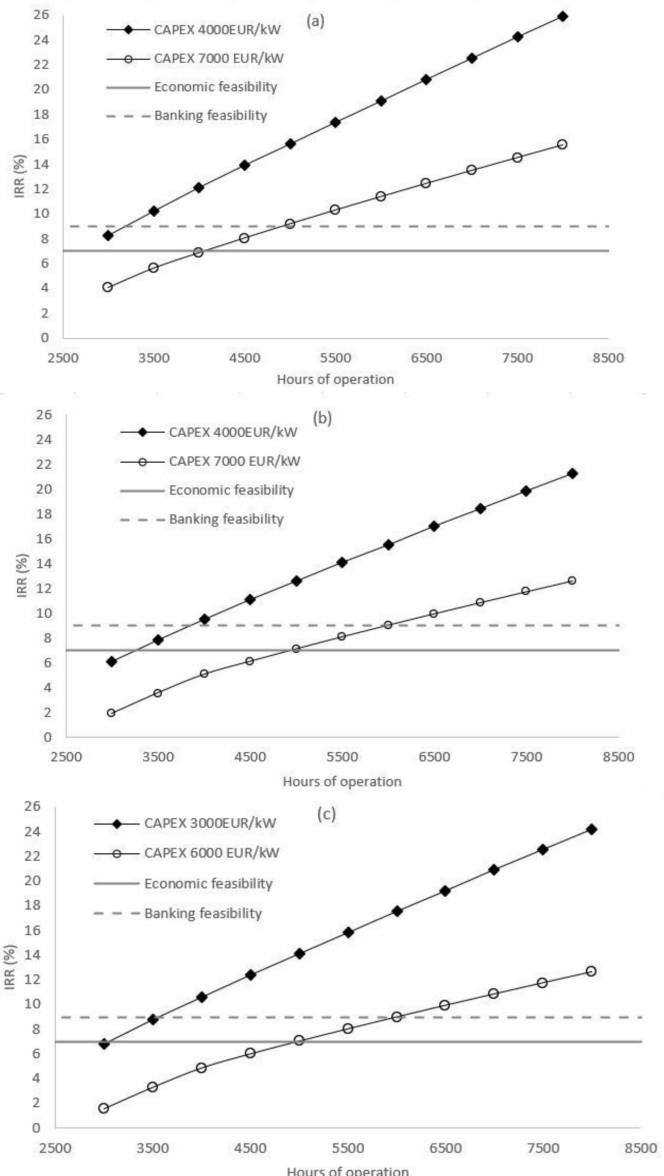

**Figure 5.** Internal rate of return vs. hours of operation for plants of 250 kW with an incentive of 0.219 EUR/kWh (**a**), 500 kW with an incentive of 0.179 EUR/kWh (**b**), 1000 kW with incentives of 0.156 EUR /kWh (**c**).

For all plant sizes, both alternatives involve advantages, except for the minimum size of 100 kW, which is characterized by low energy production that does not benefit from the extension of the PMGs or from the increase in prices. With reference to the PMG 2013, there is an increase of profitability of approximately 21%, 40%, and 46% in plants with installed power equal to 250 kW, 500 kW, and 1000 kW, respectively, while with the PMG 2019, slightly lower percentages were recorded, i.e., about 17%, 18%, and 23% for the same plant sizes.

In conclusion, the PMGs that are enforced only ensure complete profitability for the smallest plants size, i.e., with an installed power of less than or equal to 100 kW, whose OPEX is not compensated by the energy sale revenue at market price. For the other sizes, there is a deterioration in the profitability of investments as the installed power increases; this is due to the lower impact of the PMG. To make the development of new medium-large plants more attractive, an extension of the PMGs is needed.

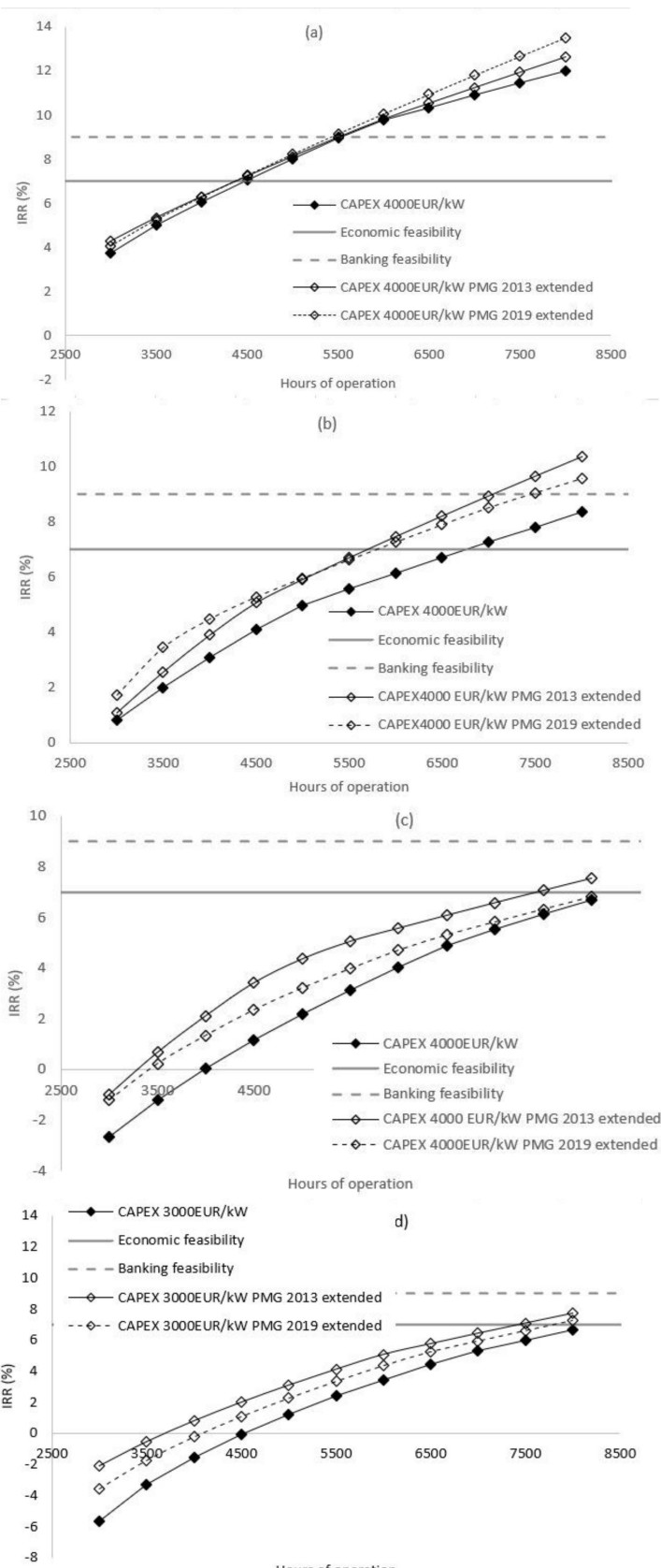

**Figure 6.** Internal rate of return vs. hours of operation comparing minimum guaranteed prices 2019 with minimum guaranteed prices 2013 extended to 2,000,000 kWh for capital expenditures 4000 EUR/kW for different plant sizes: 100 kW (**a**), 250 kW (**b**), 500 kW (**c**), and 1000 kW (**d**).

### 3.4. Phase 2—Remodulation of the Tariff Plan

The aim of this phase is to identify a so-called feed in tariff (FIT) to economically support the microhydroelectric sector in order to guarantee the maintenance of minimum plant profitability. Feed-in tariffs (FIT) are fixed electricity prices that are paid to renewable energy producers for each unit of energy produced and injected into the electricity grid. Several countries introduced this kind of energy policy [40–46]. The FIT provided by the DM 4 July 2019 was 0.080 EUR/kWh, FIT provided by DM 6 July 2012 and DM 23 June 2016 were equal to 0.155 EUR/kWh and 0.150 EUR/kWh, respectively. The same thresholds of investment acceptability and bankability as phase 1 were adopted. In this case, the cash flows were also discounted, assuming that the lifespan of the hydroelectric project is 30 years and that the energy produced was sold to the PUN. Figures 7–10 show the value of FIT to reach an IRR equal to the two pre-established thresholds (7% and 9%) for a plant size of 100 kW, 250 kW, 500 kW, and 1000 kW, with the hours of operation and the CAPEX varied.

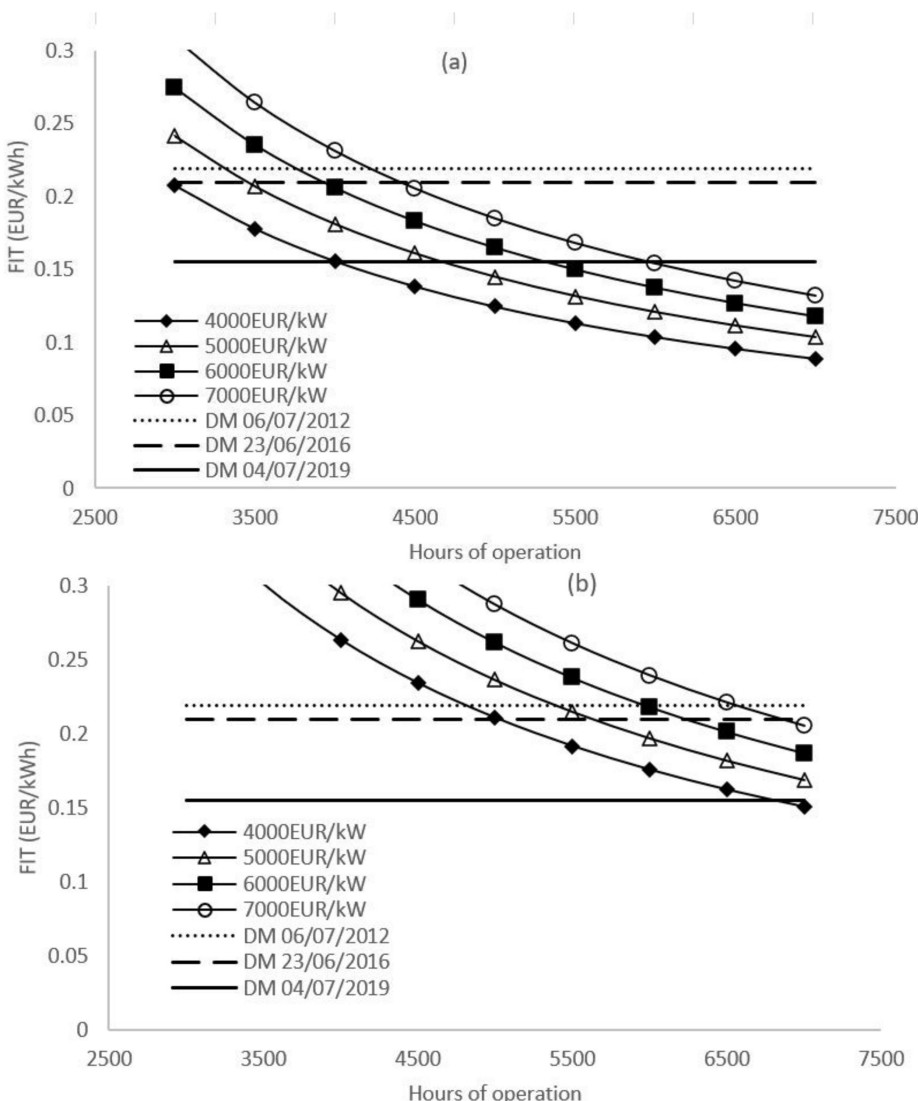

**Figure 7.** Feed in tariff vs. hours of operation for different capital expenditures to reach internal rate of return of 7% (**a**) and 9% (**b**); plant size is 100 kW.

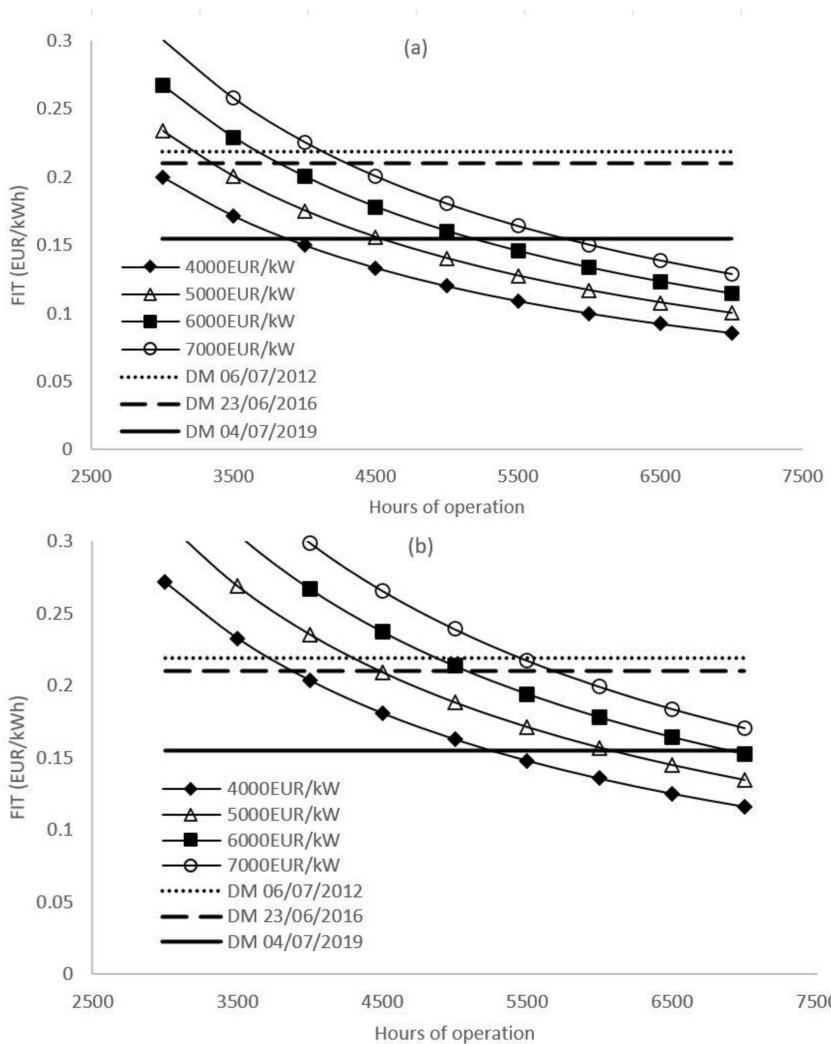

**Figure 8.** Feed in tariff vs. hours of operation for different capital expenditures to reach internal rate of return of 7% (**a**) and 9% (**b**); plant size is 250 kW.

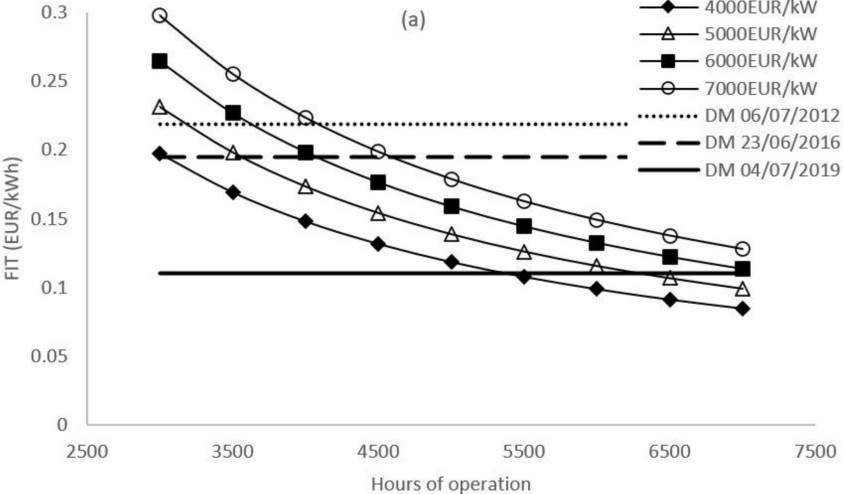

**Figure 9.** *Cont.*

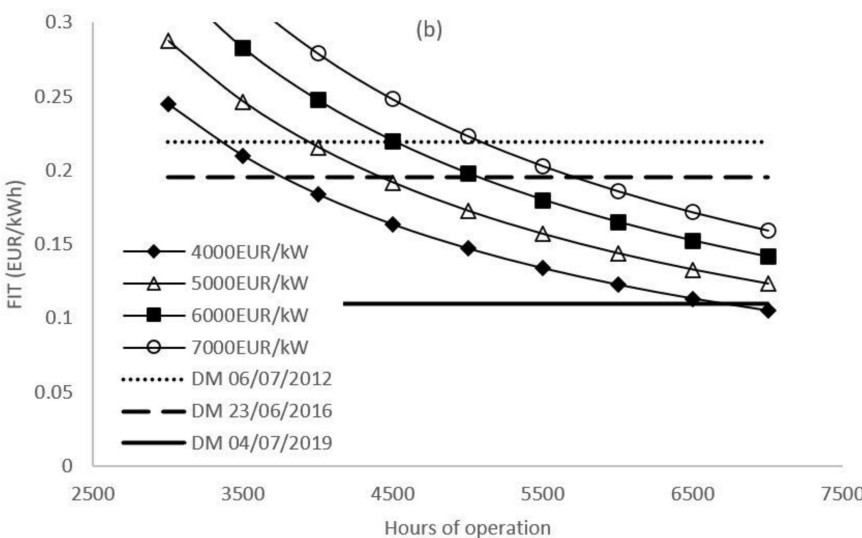

**Figure 9.** Feed in tariff vs. hours of operation for different capital expenditure values in order to reach internal rate of return of 7% (**a**) and 9% (**b**); plant size is 500 kW.

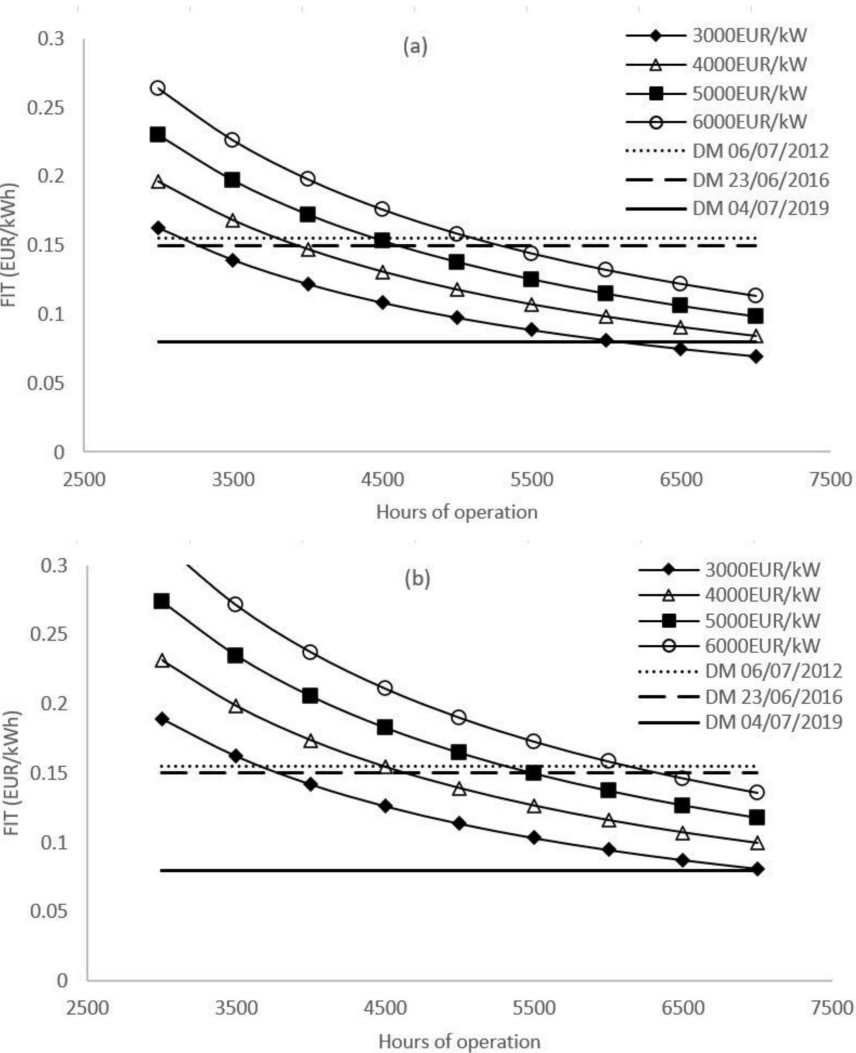

**Figure 10.** Feed in tariff vs. hours of operation for different capital expenditures to reach internal rate of return equal to 7% (**a**) and 9% (**b**); plant size is 1000 kW.

The current FIT (DM 4th July 2019) is suitable for guaranteeing a net economic return of 7% but not to reach 9%, while the FIT of DM 6 July 2012 and DM 23 June 2016 would be sufficient for both profitability thresholds. The FIT to reach an IRR of 7% and 9%, when averaging the values associated with the various CAPEX values and assuming 6000 h of operation, would be 0.129 EUR/kWh and 0.208 EUR/kWh, respectively.

Figure 8 shows some improvement for the bankability threshold: the FIT of DM 4th July 2019 (0.155 EUR/kWh) makes it possible to reach a profitability for all systems of this size with an installation cost of between 4000 and 5000 EUR/kW. Higher CAPEXs are unsustainable for this incentive rate. The FIT that would be necessary to reach an IRR of 7% and 9%, when averaging values of different CAPEX values and assuming 6000 h of operation, would be 0.125 EUR/kWh and 0.168 EUR/kWh, respectively.

With the increase in installed power, the remuneration conferred by the DM 4th July 2019 fell to 0.110 EUR/kWh with significant repercussions on the economic sustainability of a hydroelectric project (Figure 9). The minimum profitability limit is not guaranteed for an IRR of 7% (for a CAPEX of 6000 and 7000 EUR/kW) or for an IRR of 9%. The FIT to reach these profitability thresholds, when averaging values of different CAPEX values and assuming 6000 h of operation, would be 0.124 EUR/kWh and 0.155 EUR/kWh respectively.

For 1000 kW systems, the FIT provided by the DM 4 July 2019 is insufficient to guarantee an economic sustainability of 7% and 9% (Figure 10). On the contrary, the FIT established by the previous ministerial decrees, i.e., DM 6 July 2012 and DM 23 June 2016, would be sufficient for both profitability thresholds. The FIT to reach these profitability thresholds, when averaging of the values of different CAPEX and assuming 6000 h of operation, would be 0.107 EUR/kWh and 0.127 EUR/kWh, respectively. With the FIT guaranteed by the DM 23 June 2016, almost all plants reach acceptable yields at around 6000 h of operation, unlike that with the last tariff plan (DM 4 July 2019). Only systems with installed power less than or equal to 250 kW (for any CAPEX) and those of 500 kW for small CAPEX (about 4000 EUR/kW) guarantee the minimum profitability of 7% when calculating for the same number of operating hours. All other combinations do not guarantee the achievement of minimum economic sustainability. Table 5 summarizes the values of the FIT resulting from the analysis.

**Table 5.** Remodeling of the tariff plan.

| Power (kW) | FIT for 7% (EUR/MWh) | FIT for 9% (EUR/MWh) |
| --- | --- | --- |
| $1 < p \leq 100$ | 129 | 208 |
| $100 < p \leq 250$ | 125 | 168 |
| $250 < p \leq 500$ | 124 | 155 |
| $500 < p \leq 1000$ | 107 | 127 |

The current FIT (DM 4 July 2019) is insufficient for the development of plants larger than 250 kW. While in the past the incentive policies allowed for an across-the-board development of a microhydroelectric plant that affects all sizes, today they are excessively restrictive, and they risk paralyzing not only the increase in total installed power but also any technological development.

### 3.5. Phase 3—Economic Sustainability of Microhydroelectric Plants in the Period 2014–2100

The evaluation of the effect of the price of energy on production is important when studying the profitability and benefits associated with energy systems. As previously discussed, the demand and the price of electricity depend not only on economic and social developments, but they may also be subject to seasonal variability and other medium-long term variations due to climate change. With reference to the model in Figure 3, the benefit that an increase in electricity price would bring to the economic profitability of small-scale hydroelectric plants was explored. Simulations were performed by applying the *NPV* methodology; the cash flows to discount over 30 years of the investment life were updated from year to year according to the kilowatt hours of energy produced and the average

annual price. Two different analyses were carried out: the first compares the performance of hydroelectric projects undertaken in two opposing scenarios, the second investigates the evolution of the IRR from 2020 to 2070.

3.5.1. Analysis 1: Comparison of Performance for Two Opposite Scenarios

For each plant size, the trend of the IRR for two opposite scenarios, characterized by regimes of electricity prices at the extremes of the pre-established temporal projection (periods 2020–2049 and 2070–2099, respectively) were compared. Consistently with previous phases, CAPEX was varied from 4000 to 7000 EUR/kW for plants with an installed power of 100 kW, 250 kW, 500 kW, and from 3000 to 6000 EUR/kW for 1000 kW systems. To consider the uncertainty linked to future changes in water concession fees from now up to 2100 in calculations, two distinct criteria were adopted:

Approach 1: Relying on the trend in the average annual energy price according to the projection in Figure 3, the *NPV* was estimated by calculating the updated cash flows from year to year according to the electricity price for the current year.

Approach 2: The calculation of the *NPV* was developed by assuming the price of electricity to be constant over time, so that all cash flows calculated for the entire useful life of the investment are equal. Figures 11 and 12 compare the results of the two approaches.

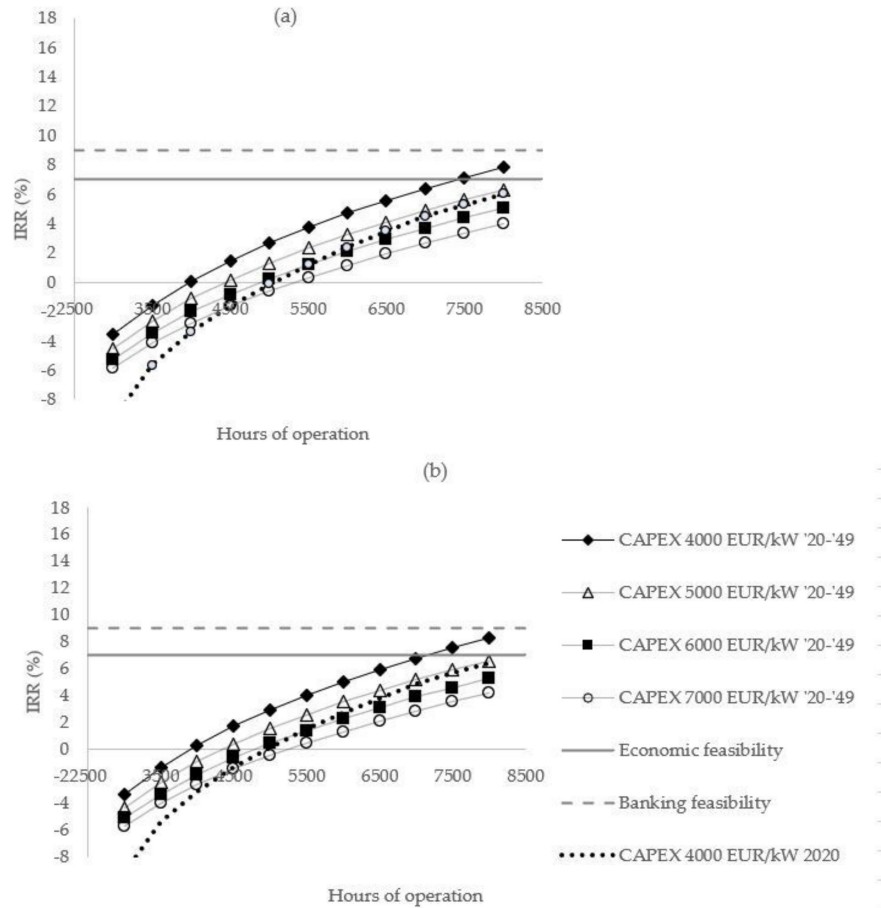

**Figure 11.** *Cont.*

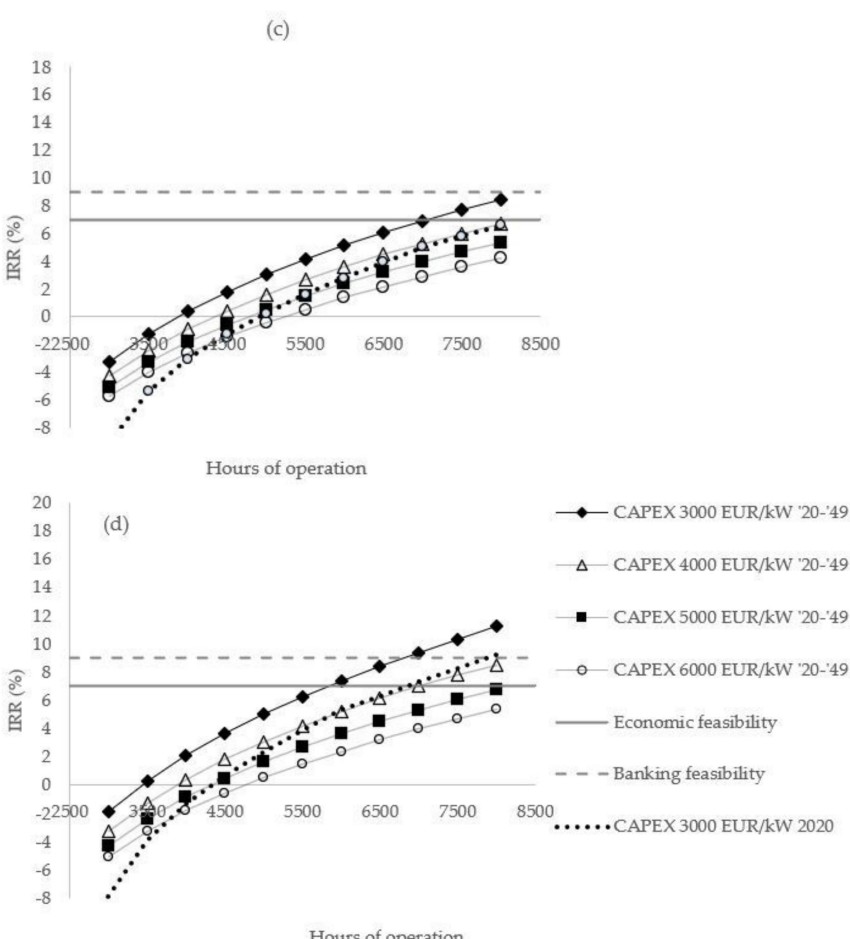

**Figure 11.** Internal rate of return vs. hours of operation, comparing scenarios 2020–2049 for Approach 1 and Approach 2: 100 kW (**a**), 250 kW (**b**), 500 kW (**c**), 1000 kW (**d**).

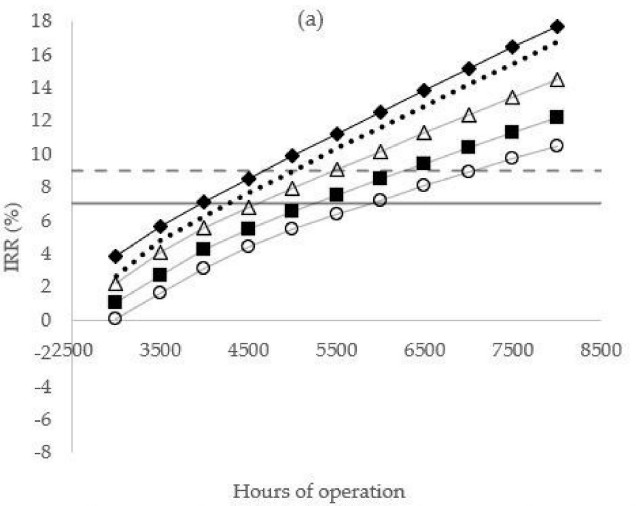

**Figure 12.** *Cont.*

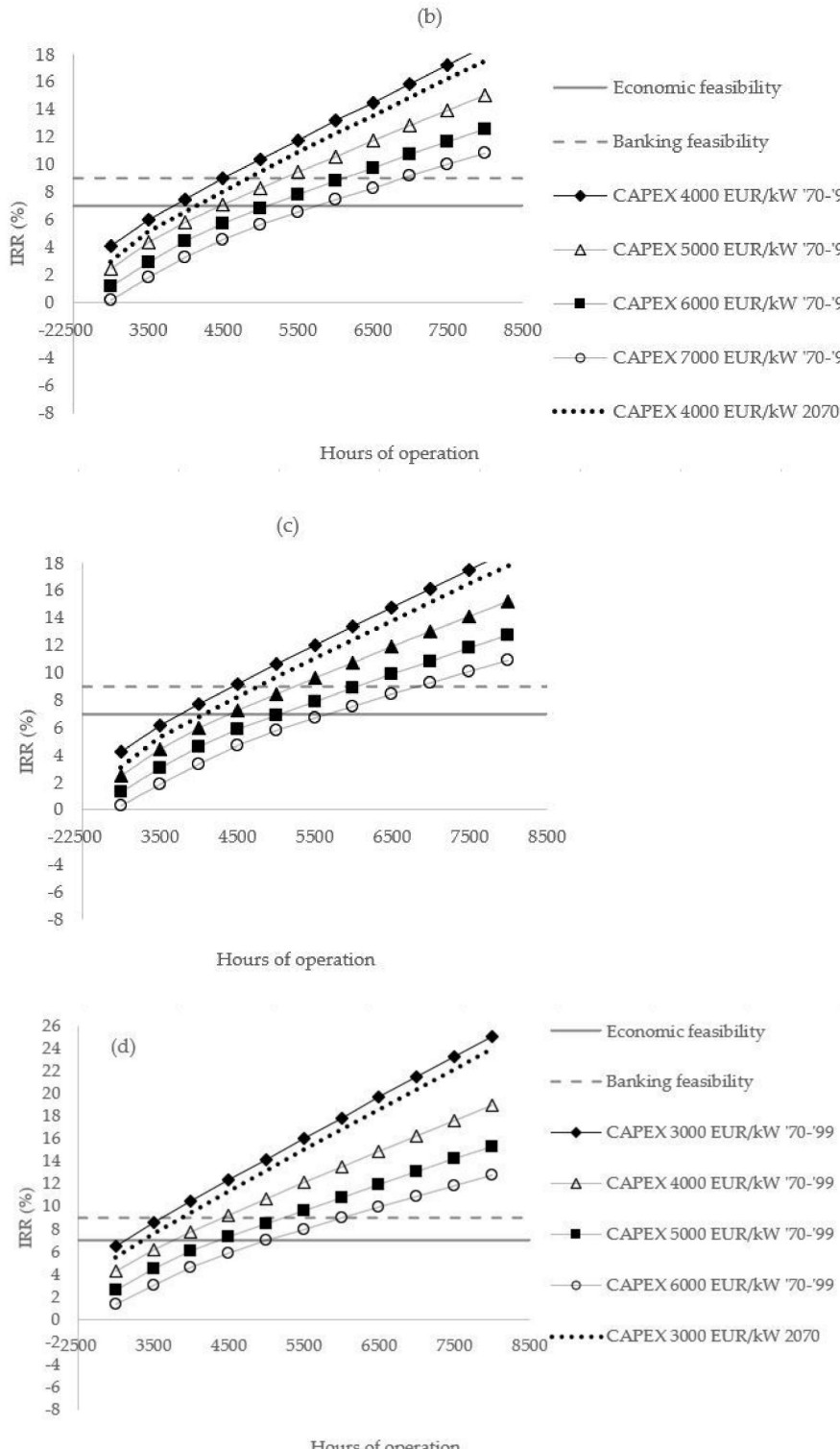

**Figure 12.** Internal rate of return vs. hours of operation, comparing scenarios 2070–2099, for Approach 1 and Approach 2: 100 kW (**a**), 250 kW (**b**), 500 kW (**c**), 1000 kW (**d**).

With reference to Approach 1, The IRR at 6000 h of operation for the period 2020–2049 never reaches profitability values equal to or greater than 7% for any CAPEX. Considering the scenario 2070–2099, the IRR always exceed the 7% threshold, up to reaching values that exceed the minimum investment convenience limit (9%) for a CAPEX below 6000 EUR/kW. The different profitability between the two stages is underlined if the situation is analyzed at 7000 average hours of operation: while in the first scenario the IRR reaches acceptable

values only for installation costs equal to 4000 EUR/kW, in the second scenario the IRR exceeds 9% for all types of investments (CAPEX of 4000, 5000, 6000, and 7000 EUR/kW). Considering a system characterized by an installed power of 1000 kW, both scenarios benefit from a more favorable IRR. The average annual production of 6000 h in the 2020–2049 scenario exceeds the minimum limit of 7% only with the minimum CAPEX, while in the 2070–2099 scenario all the IRRs record a profitability of over 9%, with values even close to 18%. The same trend occurs for 7000 average hours of production: while in Scenario 1 acceptable investments are achieved only for a CAPEX of 3000 EUR/kW and 4000 EUR/kW, in Scenario 2 all the CAPEXs exceed the convenience threshold of 9%. The positive effect on IRR of the increase in electricity prices over the course of the century is significant and reflects the expectations of the sector. On average, the increase recorded between Scenario 1 and Scenario 2, at the threshold of 6000 average hours of operation, even exceeds 100%; in other words, the profitability is doubled. Comparing results from Approach 1 and Approach 2, in all simulations the difference of IRR is appreciable. Over the useful life of a hydroelectric plant operating in the free market, the factor with the most relevant specific weight for its profitability is the price of energy. To consider the future, the evolution of the price of electricity is important for evaluating the suitability and reliability of an investment.

### 3.5.2. Analysis 2: IRR Evolution over the Period 2020–2070

The aim is to trace the trend of the investment profitability over the years, calculating the IRR by discounting the cash flows envisioned by the *NPV* methodology over the entire useful life of the system and associating the IRR to the year in which the investment is undertaken. For each year, the resulting curve indicates the percentage of the IRR of the investment, calculated by discounting the cash flows over the next 30 years (i.e., the value of profitability associated with the year 2032 is the result of the economic analysis carried out using the *NPV* method considering the cash flows in the years 2032–2061 and the corresponding energy prices). The simulations were carried out sequentially from 2020 to 2070. For each case study (i.e., 100 kW, 250 kW, 500 kW, 1000 kW), the trends of the IRR corresponding to the minimum and maximum CAPEX were calculated in order to highlight the range within which the investment profitability of similar projects can fluctuate (Figure 13).

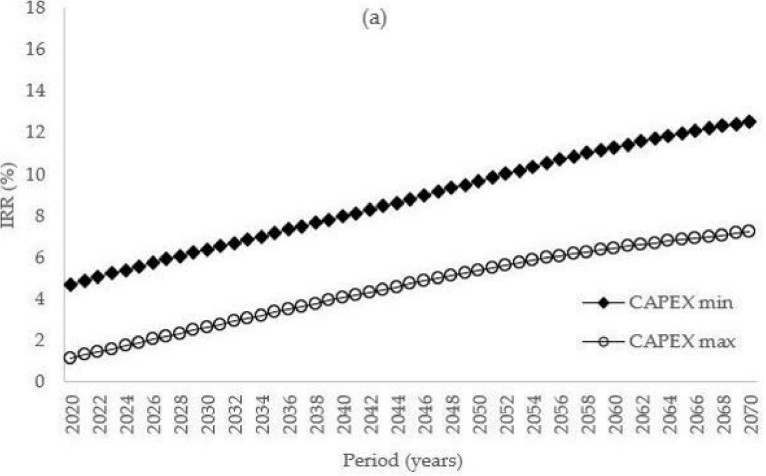

**Figure 13.** *Cont.*

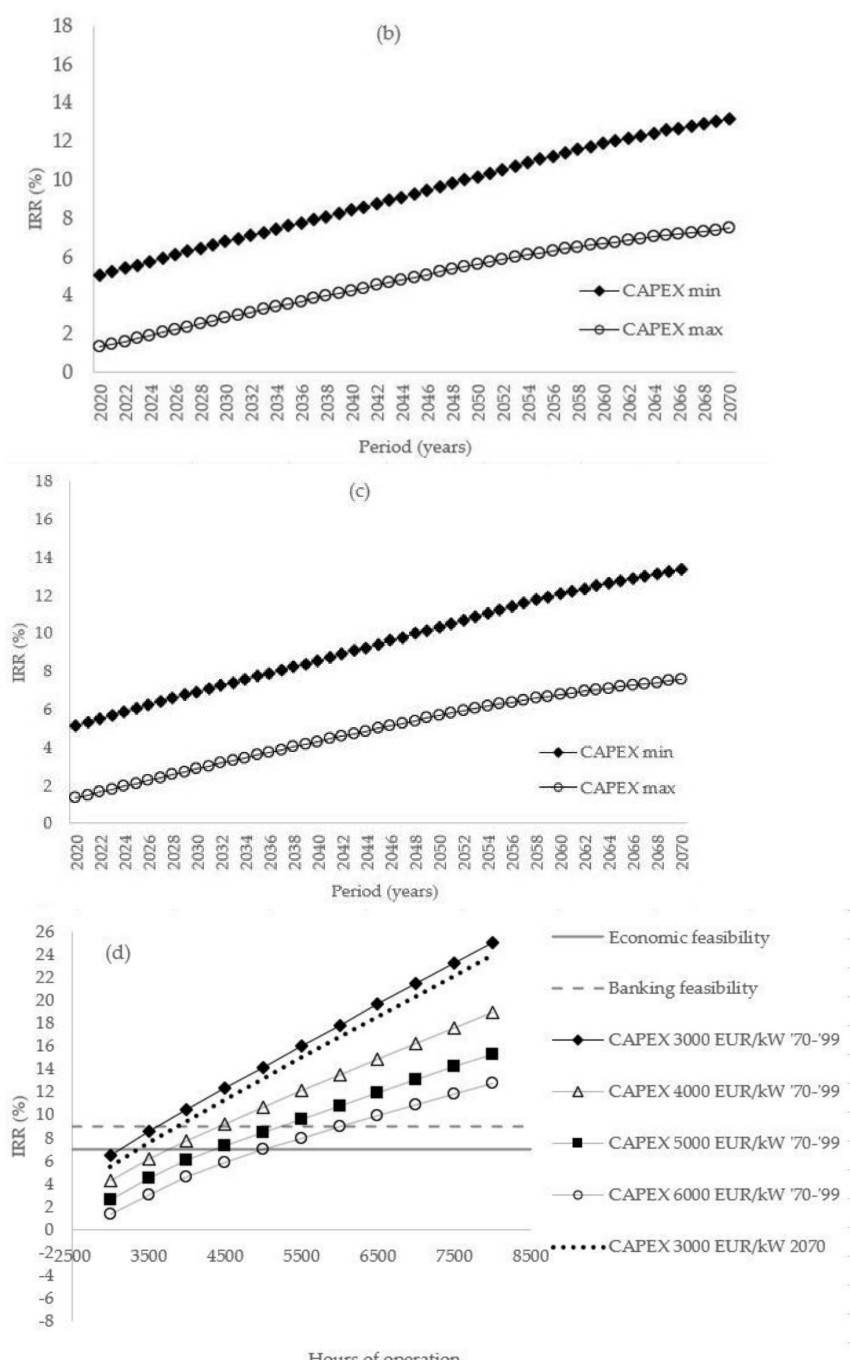

**Figure 13.** Internal rate of return vs. time in years (period 2020–2070) for capital expenditure values of 4000 EUR/kW and 7000 EUR/kW, assuming an average operation of 6000 h: 100 kW (**a**), 250 kW (**b**), 500 kW (**c**), 1000 kW (**d**).

Between one unit's installation cost and another, there is a difference ranging from 3 to 5 percentage points at the beginning of the projection (2020) and from 8 to 10 percentage points at the end of the projection (2070). The IRR increases with an almost linear trend, as expected from the trend in the price of electricity. Considering the maximum CAPEX of 7000 EUR/kW, starting from an IRR of approximately 1%, within 50 years an IRR of almost 8% is reached; considering the minimum CAPEX of 4000 EUR/kW, the initial IRR settles at around 5% and the final one at around 12.5%. The influence of the CAPEX on the marginal growth of the IRR was confirmed. Even in the case of a 1000 kW system, the IRR trend continues to increase. However, the marginal growth of this parameter is different for the two CAPEX cases considered. For the minimum CAPEX (3000 EUR/kW), the IRR

recorded in 2020 is about 7.5%, while in 2070 it settles at around 18%. For the maximum CAPEX (EUR 6000/kW), the IRR starts from a value of 1% and increases to just over 8%. In the period 2020–2070, the increase of the IRR for the maximum CAPEX is approximately 7%, and for the minimum CAPEX it is greater (about 11%). This trend can be traced back to the economic investment costs associated with the size of the plant: for the same installed power and for the maximum sizes of the microhydroelectric plant, the CAPEX assumes a greater specific weight in the projection of the IRR. For the smaller CAPEX, there is a marginal increase in the IRR, higher than the value for the same unit installation cost, for the smaller sizes. Relating the feasibility analysis of a hydroelectric project to an accurate model of future projections of the value of electricity is certainly a more truthful approach, despite the uncertainties connected with this kind of model.

## 4. Conclusions

In this study, the evolution of the profitability of an investment, according to the plant sizes and the economic context, was analyzed for the Italian scenario, as a function of the tariffs recognized by the hydroelectric energy market, the unit cost of installing the plants, and their average hours of operation. Three phases were distinguished: the first, was characterized by the analysis of the economic sustainability of the microhydroelectric plants under the PMG; in the second, the value of the incentive to reach the thresholds of "acceptability" and "bankability" of the investment, for the same hydroelectric plants as the phase 1, was estimated; in the third, an analysis of the results obtained in the previous phases was conducted using a model of the evolution of the price of electricity for the period 2014–2100. The results obtained suggest that, to maintain the attractiveness of the sector, it is necessary to safeguard access to the PMG. With PMG 2019, complete sustainability is only achieved for plants with $P \leq 100$ kW. For the remaining sizes, investments under current conditions would not be profitable. The extension of PMGs could make new medium-large plants (500–1000 kW) more attractive. The current incentive policy (DM 4 July 2019) is not effective for the development of plants larger than 250 kW; systems with lower CAPEX should be preferred. Uncertainty about the evolution of the price of energy over time is a concern for the sector; the use of evolutionary models of technical economic analysis tries to reduce these criticalities and shows that they can be transformed into opportunities. Profitability due to the growing trend expected for the price of energy cannot be highlighted by a traditional analysis.

**Author Contributions:** Conceptualization, F.B., A.B. and G.B.; methodology, F.B., A.B. and G.B.; formal analysis, A.R. and F.B.; writing—original draft preparation, A.R.; writing—review and editing, A.R., A.B. and G.B.; supervision, A.B and G.B. All authors have read and agreed to the published version of the manuscript.

**Funding:** This research received no external funding.

**Institutional Review Board Statement:** Not applicable.

**Informed Consent Statement:** Not applicable.

**Data Availability Statement:** Data sharing not applicable.

**Conflicts of Interest:** The authors declare no conflict of interest.

## Abbreviations

The following symbols are used in this paper:

| | |
|---|---|
| CAPEX | Capital expenditure |
| DM | Ministerial decree |
| FIT | Feed in tariff |
| GME | Energy markets operator |
| GSE | Energy services operator |
| IPCC | Intergovernmental panel on climate change |

| IRR | Internal rate of return |
| NPV | Net present value |
| OPEX | Operating expenses |
| PMG | Minimum guaranteed prices |
| PUN | National single price |
| RCP | Representative concentration pathway |

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
