# Peer review of "Economic Sustainability of Small-Scale Hydroelectric Plants on a National Scale—The Italian Case Study"

_water, doi:10.3390/w13091170_

Round 1
Reviewer 1 Report
This manuscript examines the feasibility of hydroelectric plants by focusing on the economic value of the produced energy.
The work is valuable of publication.
No sensible modifications are suggested but a text revision is recommended: "Analysis, referred to Italian case, were", Piedmont - Piemonte, etc.
Author Response
Dear Reviewer,
thank you for your revision to our manuscript. As, suggested we made moderate English changes by addressing to native speaker translator by a translation service.

Reviewer 2 Report
No comment for this amended edition.
Author Response
Dear Reviewer,
thank you. We are pleasure you appreciate our manuscript.
Greetings
Reviewer 3 Report
Abstract:
Soule be rewritten, and structured as follow: Background, Methods, Results, and Conclusions.
Introduction:
It provides the key elements of this section, required for scientific papers.
I suggest to create the section “Materials and methods” and move the last paragraph from “Introduction” to this new section – page 3, lines 113-125.
- Feasibility criteria:
Should be included in the section “Materials and methods”.
Eq. (1) should have source.
- Case study:
Part of this description should be moved to the section “Materials and methods”.
For the Figures and Tables in section “3” the sources should be given.
All abbreviations should be explained when used for the first time, as eg. “IPCC”,…, “PMG” in Tables. In the titles of Tables/ Figures there is recommended to not use abbreviations.
Overall:
The article is well-organised and written.
What are the political incentives / barriers to hydroelectric plants use in the EU and Italy?
Author Response
Dear Reviewer,
thank you for your valuable suggestions. We modified the manuscript, as described in the following.
- Comment: “Abstract - Should be rewritten, and structured as follow: Background, Methods, Results, and Conclusions”.
Answer: we considered your structure proposal for the abstract, highlighting background (lines 9-14), methods (lines 14-19), results (lines 20-24), conclusions (24-30).
- Comment: “Introduction - It provides the key elements of this section, required for scientific papers. I suggest creating the section “Materials and methods” and move the last paragraph from “Introduction” to this new section – page 3, lines 113-125”.
Answer: we created the section “Material and methods”, moving the last paragraph from “Introduction” to this new section (lines 128-140).
- Comment: “Feasibility criteria - Should be included in the section “Materials and methods”.
Answer: we included the section “Feasibility criteria” in the section “Materials and methods” (lines 141-169).
- Comment: “Eq. (1) should have source”.
Answer: we added a source to Equation 1 (reference 22, lines 636-637) and updated the numeration of the bibliography.
- Comment: “For the Figures and Tables in section 3 the sources should be given”.
Answer: we added a source to Figures and Tables in section 3.
- Comment: “All abbreviations should be explained when used for the first time, as eg. “IPCC”,, “PMG” in Tables. In the titles of Tables/ Figures there is recommended to not use abbreviations”.
Answer: we explained all abbreviations when used for the first time and removed abbreviations in the titles of Tables/ Figures.
- Comment: “What are the political incentives/barriers to hydroelectric plants use in the EU and Italy?”
Answer: Political incentives to hydroelectric plants use in Italy were described in section 3.1 “The Italian fare system”, while the manuscript does not focus on EU political incentives.
Greetings
